# Effect of Tensile Deformation on Residual Stress of GH4169 Alloy

**DOI:** 10.3390/ma14071773

**Published:** 2021-04-03

**Authors:** Wenxiang Zhu, Fei Zhao, Sheng Yin, Yuan Liu, Ronggui Yang

**Affiliations:** 1College of Materials and Metallurgy, Guizhou University, Guiyang 550025, China; gzuwxzhu@163.com (W.Z.); yinsheng111@163.com (S.Y.); liuyuan0851@163.com (Y.L.); yronggui97@163.com (R.Y.); 2Key Laboratory for Materials Structure and Strength of Guizhou province, Guiyang 550025, China

**Keywords:** GH4169 alloy, tensile, residual stress, dislocation movement

## Abstract

In order to reduce the residual stress of the GH4169 alloy, the effect and micro-mechanism of the tensile deformation were studied. The residual stress, dislocation density, and distribution of the GH4169 alloy were analyzed by X-ray residual stress tester, X-ray diffractometer (XRD), and electron backscatter diffraction (EBSD). The results show that: with the increase of tensile deformation, the residual stress relief first increases and then decreases. When the tensile deformation is 3%, the reduction rate of residual stress reaches the maximum, which is 90%. The mechanism of residual stress relief by the tensile treatment is that the dislocation group in the alloy is activated by tensile treatment, and the dislocation distribution in the alloy is more uniform by dislocation movement, multiplication, and annihilation so that the residual stress can be eliminated.

## 1. Introduction

The nickel-based superalloy GH4169 alloy has become a key material for aerospace applications due to its excellent comprehensive properties, and it is used to manufacture key components such as rocket engines and aeroengines [1,2,3]. The GH4169 alloy rings and thin-walled parts needed for manufacturing these components can exhibit a large deformation of residual stress due to uneven plastic deformation and temperature change during the hot working process, which will seriously reduce the dimensional accuracy of the parts, resulting in a large deviation between the size of the parts produced by hot working and the required size. To make the size of superalloy parts reach the target accuracy, it is necessary to increase the machining allowances during the hot working process. After the hot working is completed, the allowances are removed by cutting to obtain the required part size. However, cutting the machining allowance will destroy the streamline of the workpiece, which will reduce the fatigue strength of the workpiece, affect its service life and service performance, and even possibly lead to serious accidents. In addition, large machining allowances will also cause considerable material waste and increase cutting costs, which is not conducive to containing production costs. In general, due to the uneven plastic deformation and temperature changes that occur during the hot working process, it is easy to form residual stress in superalloy parts that will increase their production costs, reduce their service performance, and affect the overall performance improvements of the engine and other parts. Therefore, reducing the residual stress in GH4169 alloy workpiece has become an urgent problem to be solved by the aerospace component producers and engineers.

At present, the common methods for reducing metal residual stress mainly include the thermal aging method, vibratory stress relief (VSR), and alternating magnetic field treatment method. The thermal aging method has an obvious effect on reducing residual stress, but the method has a long process cycle and consumes a large deformation of energy, and the mechanical properties of the material are significantly reduced after thermal aging treatment [4,5,6]. The process cycle of VSR is short, and the energy consumption is low, but the effect of residual stress reduction is not ideal and can only be reduced by approximately 40% [6,7,8]. The residual stress reduction effect of the alternating magnetic field treatment method is even worse with a reduction of approximately 20% [9,10,11]. Compared with these methods, the tensile method can have both of their advantages and can compensate for their shortcomings. When using the tensile method to reduce the metal residual stress, the process cycle is extremely short, the energy consumption is very low, and the residual stress reduction effect is also significant and does not affect the material mechanical properties [12,13,14].

Yang et al. [15] found that when the tensile deformation is 1.5%, the residual stress of A357 aluminum alloy can be reduced by 94.9%. Koҫ et al. [16] found that when the tensile deformation is 2.0%, the residual stress of 7075 aluminum alloy can be reduced by 90%. Tanner et al. [17] found that applying 2% tensile deformation to a 7010 aluminum alloy thick plate can eliminate most of the residual stress. Toribio et al. [18] found that the residual stress of cold-drawn prestressing steel wires can be reduced by 90% undergoing overloads during cyclic loading, and the cause of residual stress reduction is the local constriction exerted by the plastic strain zone, which impedes the total elastic strain recovery during fatigue. The residual stress can be reduced by a certain tensile deformation. However, the current research on reducing residual stress by this method has mainly concentrated on aluminum alloy, and research on the residual stress reduction of GH4169 alloy by this method has rarely been reported. It is unknown whether GH4169 alloy can be treated with the same tensile method as aluminum alloy to reduce residual stress and achieve a better reduction effect. Moreover, the microscopic mechanism of the tensile method to reduce the residual stress of the metal is still unclear.

To reveal the effect and micro mechanism of residual stress reduction of GH4169 alloy by the tensile method, this work used an X-ray residual stress tester, X-ray diffraction (XRD), and electron backscatter diffraction (EBSD) to analyze the residual stress, dislocation density and dislocation distribution of GH4169 alloy before and after tensile deformation. This provides a simple and practical method and theoretical guidance for GH4169 alloys to reduce residual stress.

## 2. Materials and Methods

The chemical composition of the GH4169 alloy selected in this test is shown in Table 1, and the processing size of the sample is 100 mm × 50 mm × 20 mm. After the sample was heat-treated at 950 °C in an MXL (Z) -05 electric box furnace (Risine, Hefei, China) for 1 h, it was cooled with water to room temperature. The GH4169 alloy sample with residual stress was cut into several tensile specimens, as shown in Figure 1, by using electric spark. The thickness of the tensile specimen was 2.5 mm. The tensile test was carried out on an MTS 810 tensile tester, in which the loading rate was 1 mm/min, and the tensile clamping force was 120 KN. The tensile deformations selected for the test were 1%, 2%, 3%, 4%, and 5%, and the test was repeated five times for accuracy. Tensile deformation occurs in the plastic deformation zone. When the specified tensile deformation was reached, the specimen was unloaded immediately. The tests were independent of each other and each tensile specimen corresponded to only one tensile deformation. Figure 2 shows the stress-strain curve of the GH4169 alloy, and the value maximum strain is 54.59%. A STRESS-X X-ray residual stress tester produced by GNR in Italy was used to measure the residual stress of the GH4169 alloy before and after tensile deformation. Five points in the middle 10 mm × 10 mm area of the superalloy tensile specimen were selected for measurement. The average of the measurement results of five points was taken as the residual stress of the tested sample. Each point measured the X and Y directions; the X-direction meant parallel to the tensile direction, and the Y-direction referred to the direction perpendicular to the tensile direction. The specific measurement positions are shown in Figure 1, and the test parameters are shown in Table 2.

The dislocation density of the GH4169 alloy before and after tensile treatment was measured by Empyrean XRD. The scanning rate was 2 °/min, the scanning range was 30° to 100°, the rated voltage was 60 kV, the rated current was 100 mA, the target material was Cu, and the X-ray wavelength was 1.540598 nm. An AZtecHKL-EBSD (OXIG, Abingd on, United Kingdom) system, equipped on a SUPER40 (Zeiss, Oberkochen, Germany) field emission scanning electron microscope (FESEM), was used to observe the samples before and after the tensile treatment. A sample without tensile treatment was placed into the vacuum chamber of an Auger spectrometer, and FESEM was used to select a scanning micro area in the middle of this sample. The scanning area was 290 μm × 210 μm, and the scanning step was set at 0.5 μm for each point. Then, EBSD scanning was carried out, and the data were recorded. The sample after tensile treatment was scanned by EBSD at the same step length, and the data were recorded. The results were analyzed after all operations were completed. The samples needed to be electropolished before EBSD scanning. The composition of the polishing liquid was HClO_4_ and CH_3_COOH (Supelco, Beijing, China), and the volume fraction ratio was 1:9. Electropolishing was carried out at room temperature, the current was set to 0.3 A, and the polishing time was 40 s.

## 3. Results

### 3.1. Residual Stress

The residual stress test results of the GH4169 alloy before and after the tensile treatment are shown in Figure 3. The black curve indicates the residual stress in the superalloy before the tensile treatment, and the red curve represents the residual stress after the tensile treatment. As shown in Figure 3a,b the residual stresses of the sample along the X and Y directions after the tensile treatment were significantly reduced compared with those before the tensile treatment, and with the increase of the tensile deformation, the change trend of the residual stress of the sample along the X direction and the Y direction was the same. The tensile deformation was increased to 3%, and the residual stress was reduced from approximately −450 MPa to approximately −50 MPa. The residual stress was significantly reduced in this tensile interval. When the tensile deformation was increased from 3% to 5%, the magnitude of residual stress did not change significantly. From the variation trend of the residual stress of the GH4169 alloy after tensile treatment, it can be seen that the residual compressive stress can be effectively eliminated under an appropriate tensile deformation.

The reduction rates of the X-direction and Y-direction residual stresses are shown in Figure 4a,b $\sigma_{b}$ and $\sigma_{a}$ represent the average residual stress of the GH4169 alloy before and after tensile treatment, respectively. The equation of the residual stress reduction rate can be given as:(1)$$\frac{\sigma_{b}-\sigma_{a}}{\sigma_{b}}\;\times \;100\%$$

Figure 4a,b show that, with increasing tensile deformation, the reduction rate of the residual stress of the GH4169 alloy first increased and then decreased. When the tensile deformation reached 3%, the residual stress reduction rate of the superalloy along the tensile direction and perpendicular to the tensile direction reached the highest value. The residual stress reduction rate along the tensile direction reached 87%, and the reduction rate perpendicular to the tensile direction reached 91%. The tensile method has a significant effect on reducing the residual stress of the GH4169 alloy.

### 3.2. Statistics of Dislocation Density of GH4169 Alloy before and after Tensile Treatment Based on XRD

The XRD patterns of the GH4169 alloy sample, before and after tensile deformation, are shown in Figure 5. The Dunn method was used to calculate the dislocation density of the superalloy before and after tensile deformation. The change in material dislocation density will cause a change in the full width at half maximum (FWHM) of the X-ray diffraction peak. Dunn et al. [19] found that dislocation density is closely related to FWHM. The specific equation can be given as:(2)$$\rho \;=\frac{\beta^{2}}{2\mathrm{In}2b^{2}\;\;\;}$$

In the equation, β is the FWHM of the diffraction peak in Figure 5, and the FWHM of each diffraction peak can be extracted by MDI Jade 6.0 software. b is the module of Burgers vector. The GH4169 alloy has a face-centered cubic structure, and the mode of the Burgers vector is approximately 0.2546 nm.

The crystal plane of the GH4169 alloy sample was analyzed by XRD. The four crystal planes (111), (200), (220), and (311) were the crystal planes with obvious diffraction. The intensity of diffraction peaks corresponding to other crystal planes was very weak and the change was not very obvious. Thus, this work mainly focused on the four crystal planes (111), (200), (220), and (311) for analysis and research. Figure 4a,b show that the residual stress reduction effect of the GH4169 alloy is best when the tensile deformation is 3%. Therefore, this work mainly analyzes the dislocation density of GH4169 alloy with 0% and 3% tensile deformation. Jade 6.0 software (MDI, Burbank, CA, USA) was used to extract the FWHM of the diffraction peak corresponding to each crystal plane, substituting it into Equation (2) to find the dislocation density of the corresponding crystal plane, and define the equation of the dislocation density change rate:(3)$$\frac{Dislocation\;density\;after\;tensile-Dislocation\;density\;before\;tensile}{Dislocation\;density\;before\;tensile}\;\times \;100\%$$

The dislocation density of each crystal plane of the GH4169 alloy before and after tensile treatment is shown in Figure 6, and the total dislocation density is the sum of the dislocation densities of the (111), (200), (220), and (311) crystal planes. At a 0% tensile deformation, the dislocation density of the superalloy was 1.03311 × 10^15^ m^−2^, and at a 3% tensile deformation, the dislocation density was 1.08544 × 10^15^ m^−2^. According to Equation (3), the dislocation density of the superalloy increased by 5.1%. Figure 6 shows that the dislocation density of the (111) crystal plane and (311) crystal plane of the GH4169 alloy increased after tensile treatment, while that of the (200) crystal plane and (220) crystal plane decreased. From this, it can be inferred that dislocation movement, proliferation, and annihilation of the GH4169 alloy occurred during the tensile process. During the movement of the dislocation, the positive and negative dislocations met, resulting in annihilation of the dislocation. Dislocation movement will also encounter obstacles, forming dislocation loops and generating new dislocations continuously, causing the proliferation of dislocations [20]. The process of dislocation proliferation is dominant, which leads to an increase in the dislocation density of the superalloy after tensile treatment.

### 3.3. Dislocation Distribution and Grain Boundary Angle Results of GH4169 Alloy before and after Tensile Treatment Based on EBSD

#### 3.3.1. Dislocation Distribution Based on KAM

Kernel Average Misorientation (KAM) is a means used to characterize the average misorientation of an intracrystalline core that is often used to characterize the dislocation distribution in the intracrystalline region of materials [21,22]. KAM values range from 0° to 5° and are distinguished by color. Blue indicates that the KAM value is 0°, and red indicates that the KAM value is 5°. The larger the KAM value is, the greater the dislocation density.

Figure 7a,b how the KAM distribution of the GH4169 alloy sample when the tensile deformation was 0% and 3%, respectively. After the tensile treatment, the KAM distribution in the intragranular region of the GH4169 alloy changed significantly. Comparing the boxed areas in Figure 7a,b, it can be seen that the dislocations of the GH4169 alloy before the tensile treatment were mainly concentrated at the grain boundaries, the dislocation density at the grain boundaries was significantly greater than that in the intragranular region, and the distribution of dislocations was uneven. However, after the tensile treatment, the dislocations at the grain boundaries of the GH4169 alloy began to diffuse into the intragranular region and the dislocation density gap between the grain boundaries and the intragranular region gradually narrowed. The change in the KAM distribution represents the change in the dislocation distribution. In general, the distribution of dislocations showed a uniform trend.

#### 3.3.2. Small-Angle Grain Boundaries Change

The small-angle grain boundary data of the observation area of the sample were extracted by the channel5 software, and the graph and histogram were drawn as shown in Figure 8a,b and the value range of the small-angle grain boundary was 2°–10°. Fraction in the figures represents the proportion of each grain boundary angle in all grain boundary angle distributions. The small-angle grain boundaries were reduced after the GH4169 alloy sample is stretched, especially in the grain boundary angle range of 3°–7°. The small-angle grain boundaries were formed by the accumulation of dislocations, and the change in the small-angle grain boundaries largely dependson the movement of the dislocations [23]. During the tensile process of the GH4169 alloy sample, the movement, crossing, and annihilation of the dislocations packed at the grain boundaries caused a reduction in the small-angle grain boundaries. Combined with Figure 7a,b, the movement of dislocations at the grain boundaries also caused changes in the distribution of dislocations within the grains. The dislocations packed at the grain boundaries moved into the grains and generated an increase in dislocations, which increased the number of dislocations in the grains. Finally, the dislocation distribution of the GH4169 alloy became more uniform.

## 4. Discussion

The analysis results based on XRD and EBSD show that, after the GH4169 alloy was stretched, the dislocations in the alloy move. The critical stress required for the movement of the dislocations is shown in Equation (4).
τ_c_ ≈ 2 Gb/L(4)
where G is the shear modulus, b is the Burgers vector, and L is the length of the dislocation line. Taking G = 79 GPa, b = 2.5 × 10^−10^ m. The value of L was close to the grain length, L ≈ 10^−5^ m. We substituted the above data into Equation (4) to calculate τc ≈ 4 MPa. Judging from the magnitude of the critical stress, the residual stress in the GH4169 alloy can provide the driving force for dislocation movement. However, dislocations cannot move in the size of the complete grain length. They will be split into tens or even hundreds of parts by impurity atoms, vacancies, and pinning points at the grain boundaries. From Equation (4), it can be seen that the critical stress required for dislocation movement increased by tens or hundreds of times. At this time, the residual stress in the GH4169 alloy was not enough to provide the driving force for dislocation movement. When the residual stress of the GH4169 alloy was reduced by the tensile method, the external force applied to the superalloy was superimposed with the residual stress in the alloy. When the superimposed stress exceeded the driving force required for the movement of the dislocation, the dislocation started to move. The driving force required for dislocation movement is shown in Equation (5), where *τ_r_* is the driving force of the dislocation movement provided by the residual stress, and τ_s_ is the driving force of the dislocation movement provided by the tensile process.
τ_r_ + τ_s_ > τ_c_(5)

From an energy point of view, a certain potential energy barrier must be overcome if the dislocations want to continue to slip. When the activation energy of dislocations is enough to overcome the barrier energy, the slip motion of dislocations can be realized [24]. Assuming that the energy required for the dislocation to slip from one position to another is ΔF, which is also called Helmholtz free energy, this energy is provided by the mechanical energy ΔW and the thermal activation energy ΔG. An equation for ΔG can be given as [25]:ΔF = ΔW + ΔG(6)

The GH4169 alloy sample exhibited no obvious temperature change during the tensile process, and the contribution of the thermal activation energy ΔG to the dislocation movement could be ignored [25]. Therefore, the energy for activating the dislocation movement was provided by mechanical energy. The sample must reach a certain point of deformation during the tensile process to provide enough mechanical energy to activate the movement of the dislocation.

In the process of dislocation movement, meeting the positive dislocation and the negative dislocation will produce annihilation of the dislocation and cause a reduction in the dislocation density. The equation for dislocation density reduction can be given as [25,26]:(dρ^−^)/dτ = k_1_ρ(7)

Based on the Frank–Read dislocation proliferation mechanism, dislocation loops are formed when the two ends of dislocations are blocked during the movement of the dislocation, resulting in the proliferation of dislocations and an increase in dislocation density. The equation of dislocation density increase can be given as [25,26]:(dρ^+^)/dτ = k_2_ρ^1/2^(8)

By combining Equations (7) and (8), the evolution equation of dislocation density is obtained:dρ/dτ = k_1_ρ + k_2_ρ^1/2^(9)

In the equation, k_1_ is the dynamic recovery coefficient, which is related to temperature and external force, and the dislocation storage coefficient k_2_ is regarded as a constant value. The temperature of the GH4169 alloy did not change significantly during the tensile process, so the evolution of dislocations was only related to the external force that the tensile treatment acted on the material.

Figure 9 shows the external force of the GH4169 alloy under different tensile deformation. According to Figure 8 and Equations (5) and (6), a 3% tensile deformation could provide the driving force and activation energy required for the movement of GH4169 alloy dislocations so that the dislocations at the grain boundaries move. Combining Figure 7, Figure 8 and Equation (9), it can be seen that the dislocation annihilation process at the grain boundaries of the GH4169 alloy under 3% tensile was dominant, and the dislocation density at the grain boundaries decreased. In the grains, the dislocation increment process dominated, and the dislocation density in the grains increased. Eventually, the dislocation density gap between the grain boundaries and the grains was reduced. Therefore, after a 3% tensile treatment, the dislocation distribution in the GH4169 alloy tended to be uniform as a whole, and the residual stress in the superalloy was relaxed. When the tensile deformation was greater than 3%, the external force loaded in the alloy continued to increase, and the effect of reducing the residual stress was weakened when the external stress was too high [17,27].

The microscopic mechanism of reducing the residual stress of metallic materials is closely related to the movement of dislocations [28,29,30,31]. Wu et al. [32] studied a low-frequency alternating magnetic field to reduce the residual stress of a 30CrMnSiA welded sample and found that the dislocation distribution of the welded sample was more uniform after magnetic treatment, which was the reason for the relaxation of the residual stress of the sample. Xu et al. [33] studied the residual stress reduction of TC4 titanium alloy by a pulsed magnetic field and found that the dislocation density of titanium alloy increased after pulsed magnetic field treatment, and the residual stress was reduced due to the movement and uniform distribution of dislocations. This paper used XRD, EBSD, and other characterization methods to characterize the dislocations of GH4169 alloy before and after tensile treatment and obtained the same results as the above research. After the GH4169 alloy was stretched, the dislocations originally packed in the superalloy could move, and the dislocation distribution tended to be uniform, so the residual stress in the superalloy was reduced.

## 5. Conclusions

The aim of the present study was to explore the effect and micro-mechanism of the tensile method to reduce the residual stress of GH4169 alloy. To this end, the residual stress, dislocation density, and distribution of the GH4169 alloy were analyzed by X-ray residual stress tester, XRD, and EBSD. Based on the results and analysis, the following conclusions can be drawn:Tensile treatment can effectively reduce the residual stress in GH4169 alloy. When the tensile deformation is 3%, the residual stress reduction effect is the best. Under this tensile deformation, the residual stress reduction rates along the tensile direction and perpendicular to the tensile direction reached 87% and 91%, respectively;Using XRD and EBSD methods, it was observed that the dislocations in the GH4169 alloy changed significantly after tensile treatment. The dislocation density increases by approximately 5.1% after tensile treatment. The 3% tensile deformation provides a suitable driving force and activation energy for the movement of dislocations so that the dislocations packed at the grain boundaries move and annihilate, and the dislocations within the grains proliferate. The distribution of dislocations in the GH4169 alloy is more uniform, which relaxes the residual stress in the alloy.

## Figures and Tables

**Figure 1 materials-14-01773-f001:**
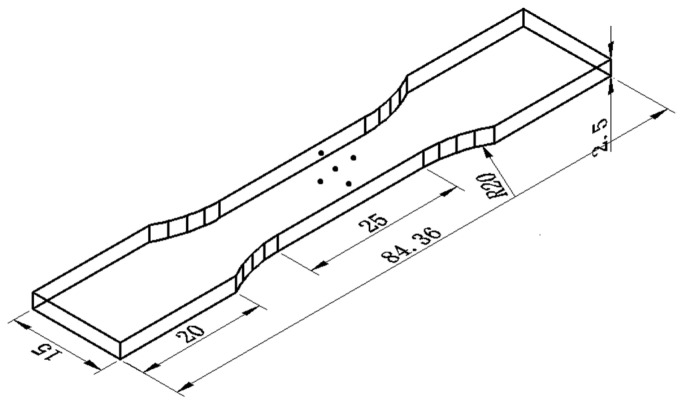
Size of tensile sample (mm).

**Figure 2 materials-14-01773-f002:**
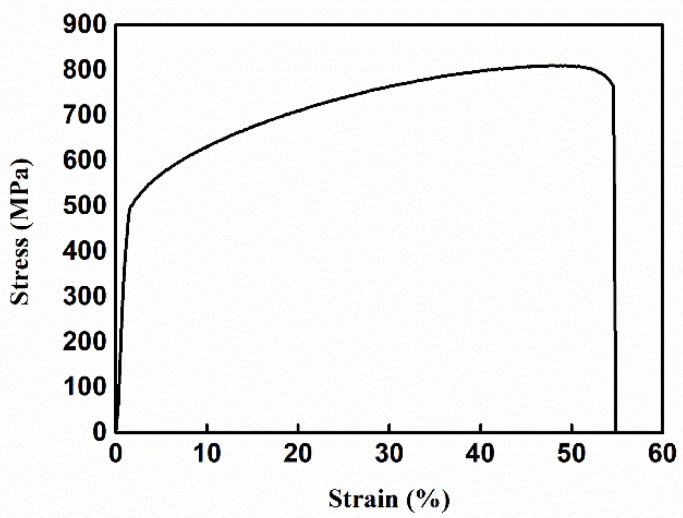
The stress-strain curve of the GH4169 alloy.

**Figure 3 materials-14-01773-f003:**
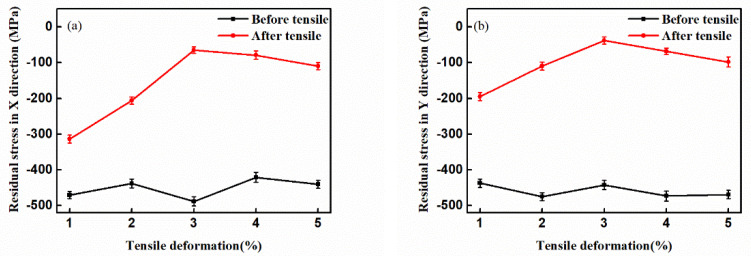
Residual stress of GH4169 alloy samples before and after tensile treatment: (**a**) Residual stress in X direction; (**b**) Residual stress in Y direction.

**Figure 4 materials-14-01773-f004:**
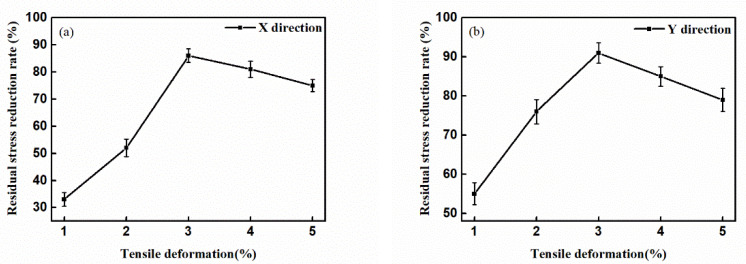
Reduction rates of residual stress at different tensile deformation: (**a**) Residual stress reduction rates in X direction; (**b**) Residual stress reduction rates in Y direction.

**Figure 5 materials-14-01773-f005:**
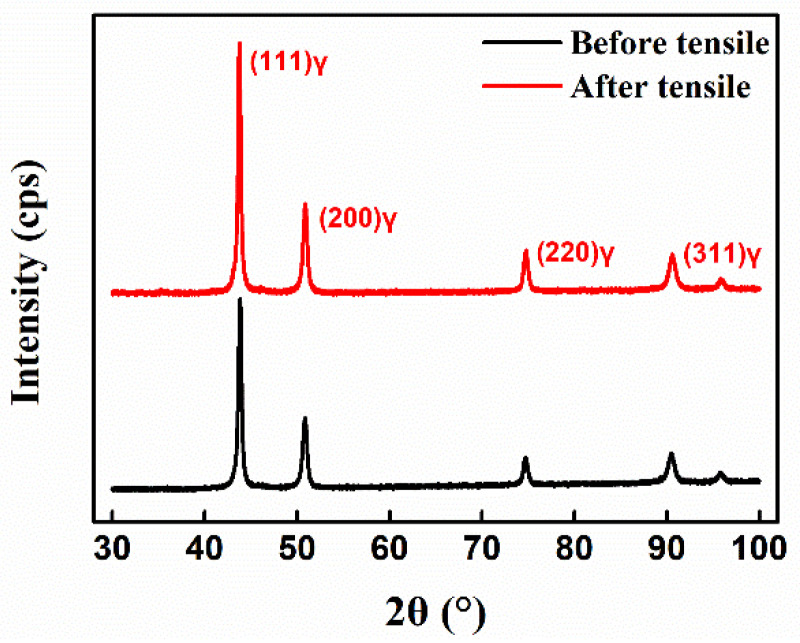
XRD patterns of GH4169 samples before and after tensile treatment.

**Figure 6 materials-14-01773-f006:**
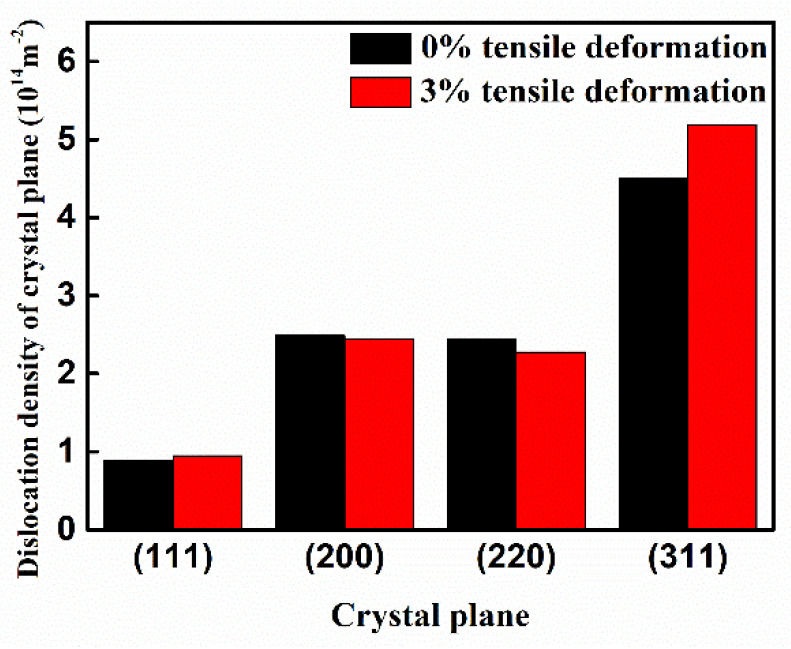
The dislocation density of each crystal plane changes before and after tensile treatment.

**Figure 7 materials-14-01773-f007:**
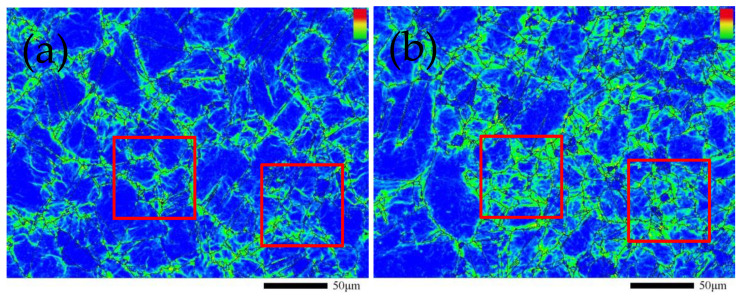
Distribution cloud map of KAM in GH4169 alloy sample before and after tensile treatment: (**a**) 0% tensile deformation; (**b**) 3% tensile deformation.

**Figure 8 materials-14-01773-f008:**
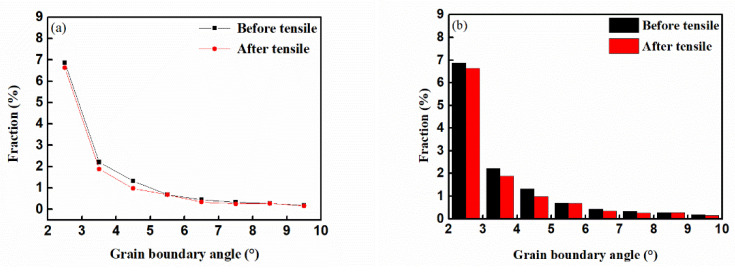
Low-angle grain boundary distributions of GH4169 alloy samples before and after tensile treatment: (**a**) Graph of grain boundary angle; (**b**) Histogram of grain boundary angle.

**Figure 9 materials-14-01773-f009:**
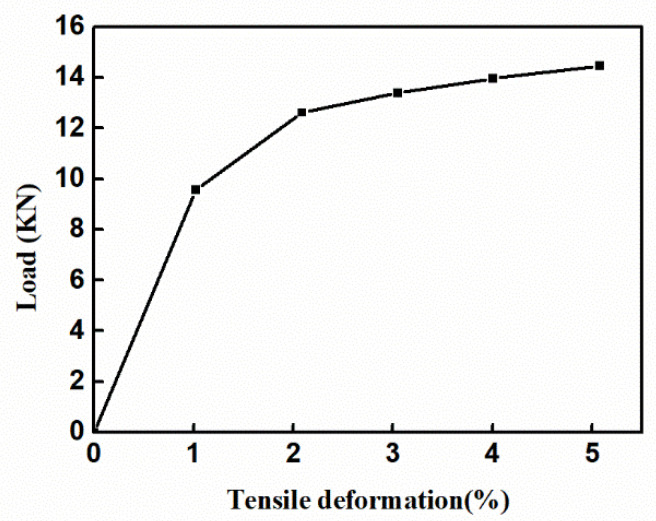
Loading force at different tensile deformation.

**Table 1 materials-14-01773-t001:** Chemical composition of the GH4169 alloy (wt.%).

**Ni**	**Cr**	**Fe**	**Nb**	**Mo**	**Ti**
52.30	19.01	17.31	5.07	3.06	1.03
**Co**	**Al**	**Mn**	**Cu**	**P**	**Bal.**
0.83	0.57	0.33	0.23	0.18	0.08

**Table 2 materials-14-01773-t002:** Relevant test parameters of residual stress tester.

**X-ray Tube Anode**	**K-β Filter**	**2-Theta Angle**	**Wavelength**
Mn K-α	Cr	152°–162°	2.1031 Å
**HKL**	**Receipt Time**	**ψ**	**Test Direction**
(311)	120s	0°, ±20°, ±40°	X&Y

## Data Availability

Not applicable.

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
