# Peer review of "Effect of Tensile Deformation on Residual Stress of GH4169 Alloy"

_materials, 2021, doi:10.3390/ma14071773_

Round 1

Reviewer 1 Report

Minor Comments:

  1. Abstract:
  • Page 1/ line 8, remove the “+”
  • Page 1/line 8 and 9, remove the repetition “to reduce …..alloy.”
  1. Introduction:
  • Page 1/ line 22, what is the name of GH4169? Is it superalloy? Add this immediately in the first sentence, please.
  • Page 2/ line 57, 83: the tensile amount of 1.5% and 1%, 2%. Is this percentage of the yield stress or altimeter stress? How the elastic stress can be used to remove the plastic stress at the microscale level (residual stress)? Can you explain and give a physical sense of this processing?
  • Page 2/line81, please change the loading speed to the loading rate.
  • Figure 1: It will be useful to show the specimen's cross-section to show the gauge thickness.
  1. Result:
  • Page 6/ line 190 and 212, remove the subtitle of 3.3.1 and 3.3.2. They are the same, and you do not need them.
  • Page 8/ line 260, add a close bracket to the reference.
  • Page 8/ line 272, Equations 8 and 9 should be Equations 8 and 7.

Reviewer 2 Report

Review for Materials- 1164321

Effect of tensile deformation on residual stress of GH4169 alloy

The authors address an interesting research topic for the journal Materials. Overall, it is a rigorous and well-organized paper. Anyway, some recommendations should be considered for its publication:

  • For further clarification of the methodology, indicate if after reaching the established % of tensile deformation the material is unloaded or remains loaded for a period of time.
  • Are the test independent of each other? I assume this is the case, but it is advisable to indicate it in the text. I
  • This paper could be related with previous papers working with short cycles of fatigue (e.g. https://doi.org/10.3390/app7010084) and further discuss the results. Please, include this aspect in the introduction section.
  • Line 81. Please, indicate in the main text that 1%, 2%, 3%, …is the tensile deformation.
  • It is advisable to include the stress-strain curve of the material and indicate the value maximum strain.
  • Please, indicate whether 1% - 5% of the tensile deformation corresponds to the elastic or plastic zone.
  • Figure 7 need to be clarified: Fraction (%)? Please, specify what percentage is.
  • Please, write the references according to the MDPI format.
  • Although the number and the selection of references could be adequate, it would be advisable to include some papers from the journals of MDPI editorial (Metals, Applied Sciences, Materials, etc.) related to the topic of the manuscript.

Round 2

Reviewer 2 Report

The paper was improved according to my previous recommendation. In my opinion, this paper can be published.

Only one more comment: Line 61, Jesus et al.... should be Toribio et al.